# Facile Preparation of Ni-Co Bimetallic Oxide/Activated Carbon Composites Using the Plasma in Liquid Process for Supercapacitor Electrode Applications

**DOI:** 10.3390/nano10010061

**Published:** 2019-12-26

**Authors:** Heon Lee, In-Soo Park, Young-Kwon Park, Kay-Hyeok An, Byung-Joo Kim, Sang-Chul Jung

**Affiliations:** 1Department of Environmental Engineering, Sunchon National University, 255 Jungang-ro, Suncheon, Jeonnam 57922, Korea; honylee@hanmail.net (H.L.); poisjs@naver.com (I.-S.P.); 2School of Environmental Engineering, University of Seoul, 163 Seoulsiripdaero, Dongdaemun-gu, Seoul 02504, Korea; catalica@uos.ac.kr; 3Department of Nano & Advanced Materials Engineering, Jeonju University, 303 Cheonjam-ro, Jeonju 55069, Korea; khandragon@jj.ac.kr; 4A Carbon Valley R&D Division, Korea Institute of Carbon Convergence Technology, 110-11 Banryong-ro, Jeonju 54853, Korea; kimbj2015@gmail.com

**Keywords:** supercapacitor electrode, Ni-Co bimetallic oxide nanoparticle, activated carbon, plasma in liquid process, specific capacitance

## Abstract

In this study, a plasma in a liquid process (PiLP) was used to facilely precipitate bimetallic nanoparticles composed of Ni and Co elements on the surface of activated carbon. The physicochemical and electrochemical properties of the fabricated composites were evaluated to examine the potential of supercapacitors as electrode materials. Nickel and cobalt ions in the aqueous reactant solution were uniformly precipitated on the AC surface as spherical nanoparticles with a size of about 100 nm by PiLP reaction. The composition of nanoparticles was determined by the molar ratio of nickel and cobalt precursors and precipitated in the form of bimetallic oxide. The electrical conductivity and specific capacitance were increased by Ni-Co bimetallic oxide nanoparticles precipitated on the AC surface. In addition, the electrochemical performance was improved by stable cycling stability and resistance reduction and showed the best performance when the molar ratios of Ni and Co precursors were the same.

## 1. Introduction

In recent years, electrochemical capacitors (ECs), called supercapacitors, have attracted considerable attention from industry and researchers as useful energy storage devices because of their long cycle life and faster charge-discharge rates than conventional batteries [1,2]. Supercapacitors can be classified as pseudo capacitors and electric double-layer capacitors (EDLCs) according to their energy storage mechanisms [3,4]. Researchers have examined pseudocapacitive transition metal oxides to improve the power and energy density of supercapacitors [5,6]. Because various transition metal oxides have high specific capacitances, they are applied as useful materials for supercapacitors [7,8]. A range of transition metal oxides has been applied to NiO [9], Co_3_O_4_ [10], RuO_2_ [11], MnO_2_ [12], and VOx [13] pseudocapacitors. On the other hand, these transition metal oxides often have poor electrical conductivity, which reduces the overall capacitance [14].

Recently, composite electrodes, in which two or more transition metal oxides are applied to an electrode material, have been proposed to solve the problem of using a single component [15,16,17]. Nanostructured composite electrodes doped with transition metal oxides in carbon-based materials showed higher specific capacitance and structural integrity after repeated cycling [18].

Recently, the synthesis of nanoparticles [19,20], production of hydrogen [21,22], and removal of organic contaminants in water [23,24] using plasma fields generated by glow discharge in water have attracted considerable research attention. Plasma in liquid process (PiLP) uses a plasma with electrons and active species to synthesize a range of nanoparticles and nano-metal oxide particle/carbon composites more easily and simply than conventional nanoparticle synthesis methods [25]. PiLP can produce single-component nanoparticles as well as multicomponent nanoparticles and composites in a single step process without the use of special additives [26].

In this study, composites prepared with transition bimetallic nanoparticles were fabricated to improve the electrochemical performance of activated carbon. Bimetallic nanoparticles were precipitated in a single step on the surface of activated carbon by PiLP using nickel and cobalt metal salt precursors. The prepared Ni-Co bimetallic oxide/activated carbon composites (NCOCC) were examined for their physicochemical properties using a range of analyzers, and the electrical properties as supercapacitor electrodes were evaluated by fabricating full-coin cells.

## 2. Experimental

### 2.1. Materials and Apparatus

YP-50F, a commercial activated carbon (AC), was purchased from Kuraray Chemical (Osaka, Japan) to prepare NCOCC using PiLP. Nickel nitrate hexahydrate (Ni(NO_3_)_2_∙6H_2_O, Sigma-Aldrich, St. Louis, MO, USA) and cobalt chloride hexahydrate (CoCl_2_∙6H_2_O, Junsei Chemicals, Tokyo, Japan) were used as precursors to precipitate Ni-Co bimetallic nanoparticles on an AC surface using PiLP. The aqueous reactant solutions mixed with the precursors and AC powder were prepared from deionized (DI) water (Daejung Chemical & metals, Siheung, Korea), wherein the electrical conductivity of DI water was less than two Siemens.

For the preparation of NCOCC, the PiL system was introduced in this study. Figure 1 presents the device configuration. A description of the experimental setup like the PiL reactor used in this experiment is described in detail elsewhere [21,22].

The power supply (NTI-1,000W) manufactured by Nano Technology Co., Ltd. (Daejeon, Korea) was designed as a bipolar pulse type to prevent corrosion of the tungsten electrode. The electrical power operating conditions applied in this experiment were as follows: pulse width of 5 μs, frequency of 30 kHz and applied voltage of 250 V, which are the optimal conditions for plasma generation. The PiL reactor, which was made of Pyrex^®^, was a double tube type with an outer diameter of 40 mm and a height of 80 mm. The aqueous reactant solution was placed in the reactor and the plasma was discharged. Cooling water circulated from a constant temperature water bath was supplied to the outer channel of the PiL reactor to maintain a temperature of 20 °C. The electrodes installed at the center of the PiL reactor were tungsten electrodes (Wolfram industrie, Traunstein, Germany) with a diameter of 2 mm and a purity of 99.95%. The electrodes were insulated with ceramic insulators and a 1.0 mm spacing was maintained where the plasma was the most stable.

### 2.2. Experimental Procedure

The procedure for synthesizing NCOCC using PiLP is as follows. Precursor nickel nitrate and cobalt chloride were completely dissolved in 250 mL of DI water at a constant molar ratio. The aqueous reactant solutions were produced by adding 0.5 g of AC and stirring for 30 min. The PiL reactor was filled with the reaction solution, and plasma was generated for 60 min by applying power to the tungsten electrode to synthesize NCOCC. Unreacted materials present in the PiL reactant solution were removed by repeated centrifugation at 10,000 rpm and washing with deionized water. Finally, NCOCC powders were obtained by filtration using a membrane filter with a pore size of 0.2 μm and drying in a 353 K vacuum oven for 48 h.

### 2.3. Characterization of NCOCC

The chemical composition and elemental distribution of NCOCCs powders were observed by EDS attached field emission scanning electron microscopy (FESEM, JSM-7100F, JEOL, Tokyo, Japan). The chemical states of the NCOCCs were measured by X-photoelectron spectroscopy (XPS, Multilab 2000 system) manufactured by Thermo Fisher Scientific, Waltham, MA, USA. Full coin cells were prepared to evaluate the electrical performance of the NCOCCs as the electrodes of supercapacitors. The slurry used in the full coin cell was mixed with super-P (TIMCAL graphite and carbon, Bironico, Switzerland) as conducting agent, carbon sources (AC or NCOCC powders), and polyvinylidene fluoride as a binder at a mass ratio of 1:8:1, and a homogeneous paste was prepared using N-methyl-2-pyrrolidone. The slurry was cast in nickel foil (20 μm, purity> 99.9%) as a current collector and then dried in an 80 °C oven for 24 h. Round electrode samples with a diameter of 12 mm were prepared using coated foils. The coin cell consisted of 150 μm glass felt between the two symmetric electrode samples and the electrode, using 1 M KOH as the electrolyte. Galvanostatic charge-discharge, cyclic voltammetry, and electrochemical impedance spectroscopy (EIS) were measured using a VSP potentiostat manufactured by Bio-logic Science Instruments, Seyssinet-Pariset, France. The electrochemical characteristics of the coin cells prepared from the NCOCC powder were evaluated from the cycling stability, current-voltage (C-V) curve, Nyquist plot, VSP potentiostat, and cyclic voltammetry.

## 3. Results and Discussion

### 3.1. Characterization of NCOCCs

The elemental composition of NCOCCs prepared using PiLP was examined by EDS attached to FE-SEM. Table 1 lists the chemical composition of NCOCCs prepared by PiLP by varying the molar ratio of the precursors of nickel and cobalt along with that of bare AC. AC (YP-50F) used as a basic raw material in this study contained 2.29 At. % oxygen. In the case of NCOCCs prepared by PiLP, the Ni, Co and O contents were higher than bare AC, and the composition of nickel and cobalt in NCOCCs was affected by the molar ratio of precursors. In NCOCC-55 with the same molar ratio of Ni precursors and Co precursors, the At. % of Ni was higher than that of Co. This is because the standard reduction potential (E°) of Ni^2+^ (−0.25 eV) is higher than that of Co^2+^ (−0.28 eV), hence, there is a difference in reducibility by a reaction with radicals generated in the plasma field [27]. In addition, the same trends were observed for the synthesized NCOCCs in the experiments using NCOCC-19 and NCOCC-91. Compared to bare AC, the O content of the NCOCCs was approximately 0.58–0.76 At. % higher, which suggests that Ni and Co precipitated on the AC surface by PiLP in the form of oxides.

The elemental distribution of NCOCCs prepared using PiLP was examined by EDS attached to FE-SEM. Figure 2 shows the NCOCC prepared using NCOACC-55 with the same molar ratio of nickel and cobalt precursors in the aqueous reactant solution. Precipitated Ni and Co oxide nanoparticles were not observed, and the images were assumed to be of AC powders. Figure 2 also shows mapping images of oxygen (pink dot), nickel (white dot), and cobalt (yellow dot) in the same region as the real image. Nickel and cobalt precipitated uniformly on AC. Ni^2+^ and Co^2+^ were produced in the aqueous reactant solution by the dissociation of nickel and cobalt precursors were reduced to Ni^0^ and Co^0^ due to chemical activated species in the plasma field, particularly hydrogen radicals, and precipitated on the AC surface [28,29]. On the other hand, the mapping result shown in the upper right corner of Figure 2 shows the presence of many oxygen elements on the AC surface. Therefore, Ni and Co precipitated as oxides on the AC surface, as predicted in Table 1. Ni and Co are oxidized by strong oxidizing species, such as oxygen atoms and hydroxyl radicals, generated in the plasma field by PiLP [30,31].

The shape of the Ni and CO nanoparticles precipitated on the AC surface could not be observed by FE-SEM. FE-TEM attached EDS was used to examine the shape and chemical composition of the Ni and Co nanoparticles on the AC surface of NCOCC, as shown in Figure 3. Figure 3a shows an image of a spherical particle, approximately 100 nm in size precipitated on the AC surface. Figure 3b,c shows elemental mapping images of the Ni (yellow dot) and Co (red dot), respectively. These images are elemental mappings of the same particle in Figure 3a. Therefore, the nanoparticles precipitated on the AC surface were found to be bimetallic nanoparticles containing both Ni and Co. Ni^2+^ and Co^2+^ ions present in the aqueous reactant solution were reduced by PiLP, which together formed the bimetallic particles. Figure 3d shows the elemental line scanning profile of the bimetallic nanoparticles in Figure 3a. The intensity of the Ni was higher than that of Co, and when bimetallic nanoparticles were formed, the reduction of Ni was preferred over Co. These results agree with those of NCOCC-55 in Table 1.

Figure 4 shows the XPS spectra of NCOCC-55. Figure 4a presents the survey spectrum of NCOCC-55. Strong peaks for C1s and O1s constituting activated carbon were noted. Weak peaks for Co2p and Ni2p were observed at 780–810 eV and 850–890 eV, respectively.

Figure 4b shows the high-resolution XPS spectra for the O1s region; three distinct peaks were observed. The peak observed at binding energy (BE) of 530.5 eV was assigned to oxygen (Ni-O-Co) coupled with Ni and Co of NCOCC. The peaks at 532.0 eV and 533.6 eV were attributed to the oxygenated species of AC (C-O and C=O) [32]. Figure 4c shows the high-resolution XPS spectra of the Ni2p region. Ni2p_3/2_, Ni2p_1/2_, and satellite peaks were observed. The peaks at 855.6 eV (Ni2p_3/2_) and 873.1 eV (Ni2p_1/2_) are two spin-orbit doublets of Ni^2+^ and Ni^3+^ [33,34]. The same trend was observed in the Co2p region, and the splitting energy of the Co2p_3/2_ peak at 781.2 eV and Co2p_1/2_ at 796.5 eV was 15.3 eV, which means that Co^2+^ and Co^3+^ cations coexist [33,34]. Overall, the Ni-Co bimetallic oxide particles that precipitated in NCOCC by PiLP are oxides of NiCo_2_O_4_, in which Ni^2+^, Ni^3+^, Co^2+^, and Co^3+^ coexist [35,36].

### 3.2. Electrochemical Performance of NCOCCs

The electrochemical performance of NCOCCs prepared via PiLP as a supercapacitor electrode was measured by cyclic voltammetry. Figure 5 shows the C-V profile of bare AC and NCOCCs measured by cyclic voltammetry. The measurement conditions were evaluated with an actuation voltage of 0.0 V to 0.8 V at a scan rate of 50 mV/s. Both bare AC and NCOCCs had rectangular C-V profiles, and no peaks generated by redoxation were observed. The capacitive currents of NCOCCs incorporating Ni-Co bimetallic oxide nanoparticles increased by more than that of bare AC, indicating an improvement in specific capacitance [37].

The Ni-Co bimetallic oxide nanoparticle precipitated on the AC surface by PiLP reacts with the electrolyte in the following reaction, which is believed to improve the specific capacitance [35].
NiCo_2_O_4_ + OH^−^ + H_2_O ↔ NiOOH + 2CoOOH + 2e^−^(1)
CoOOH + OH^−^ ↔ CoO_2_ + H_2_O + e^−^(2)

On the other hand, when looking at the effects of the precursor of the Ni and Co molar ratio, the area of the C-V profile of NCOCC-91 increased slightly compared to NCOCC-19. Table 1 shows that the total amount of Co and Ni precipitated through PiLP was slightly higher in NCOCC-91 (amount of bimetallic oxide nanoparticles, 0.28 at. %) than in NCOCC-19 (amount of bimetallic oxide nanoparticles, 0.26 at. %). In this C-V profile, the capacitance increased with an increasing amount of precipitated bimetallic oxide nanoparticles. For this reason, the large capacitance of NCOCC-91 was considered. NCOCC-55 (amount of bimetallic oxide nanoparticles 0.35 at. %) showed the highest specific capacitance because the amount precipitated was higher than that of NCOCC-19 and NCOCC-91. When the molar ratios of the Ni-Co precursors in the aqueous reactant solution were the same, the electrodes had the best reversibility and electrochemical activity, thereby improving the capacitive performance [38].

The composite resistance of the NCOCCs prepared by PiLP was compared with that of the bare AC, as shown in the Nyquist plot (Figure 6). The measurements were taken in the range of 10 mHz to 300 kHz. The semicircle, which shows in the high-frequency region, equivalent series resistance (ESR), means charge transfer resistance at the interface of the electrode material and electrolyte. Therefore, the higher the value, the higher the resistance, which negatively affects the electrochemical properties. When bare AC was used, the size of the semicircle was measured to be 4.37 Ω. However, NCOCC in which Ni-Co bimetallic oxide nanoparticles were precipitated on the AC surface by PiLP showed lower resistance due to the decrease in the size of the semicircle (NCOCC-19 (3.95 Ω), NCOCC-91 (3.44 Ω) and NCOCC-55 (3.11 Ω)). Precipitated Ni-Co bimetallic oxide nanoparticles on the surface of AC could improve the electrical conductivity of the composite, which leads to the smooth transfer of charge, thereby reducing the resistance [37]. On the other hand, the lowest ESR value was shown in NCOCC-55, which prepared bimetallic oxide nanoparticles using the same concentration of precursor. The slope of the curve in the low-frequency region indicates the resistance that occurs during ion diffusion and transport from the electrode to the electrode, and the vertical shape indicates pure capacitive behavior. Compared to pure AC, the slope of NCOCCs to which Ni-Co bimetallic oxide nanoparticles were precipitated was shifted more vertically, among which NCOCC-55 showed the most vertical slope. Bimetallic oxide nanoparticles precipitated on the AC surface by PiLP increased the ion diffusion rate and were transported smoothly. From the above results, the resistance characteristics of the composite are improved by Ni-Co bimetallic oxide nanoparticles generated on the AC surface, the best properties when the molar ratio of Ni and Co precursor is 5:5.

To evaluate the cycling stability of NCOCCs prepared by PiLP, the charge-discharge process was repeated for 5000 cycles. Figure 7 compares the change in specific capacitance with that of the capacitance of bare AC. The bare AC showed an initial specific capacitance of 24.67 F/g, which reduced to 22.50 F/g after 5000 cycles, resulting in an 8.8% capacitance loss. For the NCOCCs prepared by PiLP, the initial specific capacitances were NCOCC-55 (27.23 F/g), NCOCC-91 (26.37 F/g), and NCOCC-19 (25.84 F/g), which agrees with the result shown in Figure 5. The specific capacitance after 5000 cycles was measured as NCOCC-55 (25.48 F/g), NCOCC-91 (24.47 F/g), and NCOCC-19 (23.89 F/g), respectively, with a loss yield of 6.4% to 7.6%. This value was lower than that of the bare AC. The specific capacitance was enhanced by the reduction of Ni-Co bimetallic oxide nanoparticles precipitated by PiLP, and the electrical conductivity and reversibility were enhanced, resulting in stable charge-discharge and improved cycling stability [38]. These results confirmed that Ni-Co bimetallic oxide nanoparticles precipitated by PiLP play an important role in the electrochemical performance.

## 4. Conclusions

Bimetal nanoparticles composed of nickel and cobalt elements were easily precipitated on the surface of activated carbon using PiLP using a single process. Ni-Co bimetallic nanoparticles produced by PiLP were uniformly dispersed on the AC surface and had a spherical shape of about 100 nm. The chemical composition of the nanoparticles to be deposited was determined by the concentration ratio of precursors in aqueous solution, and the composition of Ni element was higher than that of Co element at the same precursor concentration. XPS showed that most of the nanoparticles precipitated with PiLP were NiCo_2_O_4_ with Ni^2+^, Ni^3+^, Co^2+^, and Co^3+^. Both bare AC and NCOCCs had a rectangular C-V profile and no peaks produced by redox were observed. The specific capacitances of NCOCCs in which Ni-Co bimetallic oxide nanoparticles precipitated on the AC surface were higher than those of AC powder alone. The specific capacitance of NCOCC prepared using the aqueous reactant solution with the same molar ratio of the precursors of Ni and Co was the largest. Precipitated Ni-Co bimetallic oxide particles increased the electrical conductivity and decreased the resistance of NCOCCs. The NCOCCs prepared by PiLP showed higher cycling stability than bare AC and increased with increasing precipitation of Ni-Co bimetallic oxide nanoparticles.

## Figures and Tables

**Figure 1 nanomaterials-10-00061-f001:**
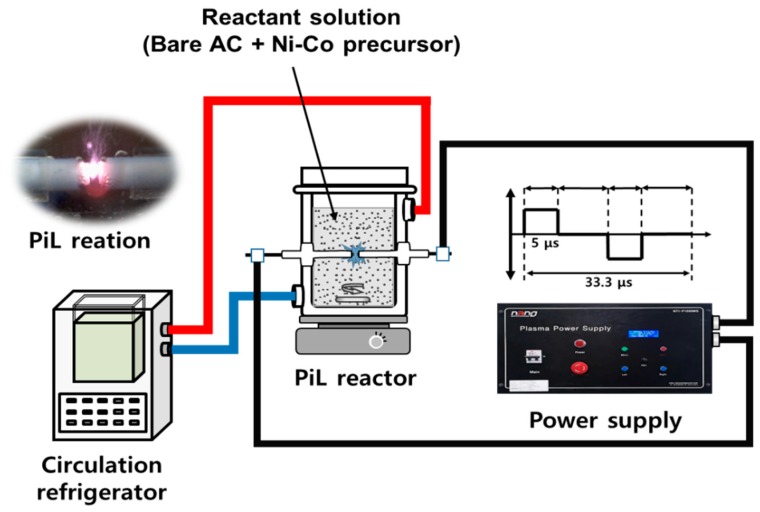
PiLP system schematic and PiL reaction photos used to prepare NCOCCs.

**Figure 2 nanomaterials-10-00061-f002:**
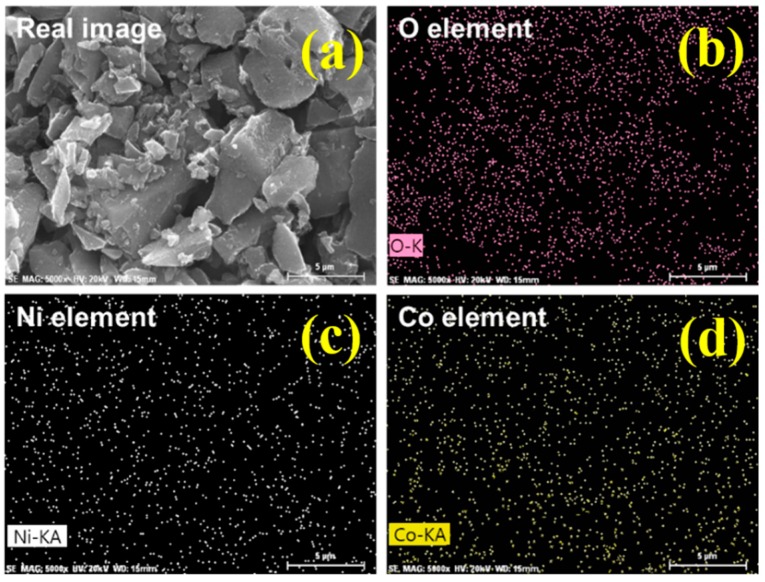
The FE-SEM real image of NCOCC-55 (**a**) and the oxygen (**b**), Ni (**c**) and Co (**d**) mapped by EDS analysis.

**Figure 3 nanomaterials-10-00061-f003:**
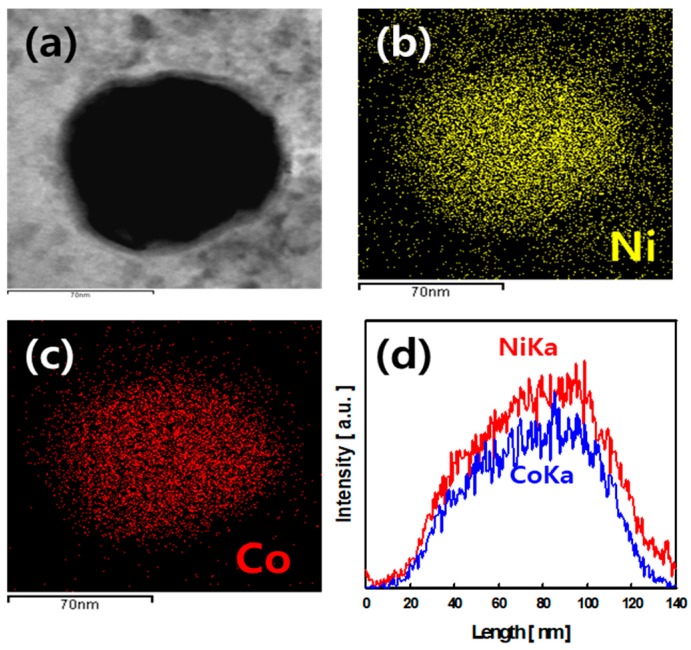
Bimetallic particle analysis of NCOCC-55 using FE-TEM and EDS. FE-TEM real image (**a**), Ni elemental mapping (**b**), Co elemental mapping (**c**) and line scanning profile (**d**).

**Figure 4 nanomaterials-10-00061-f004:**
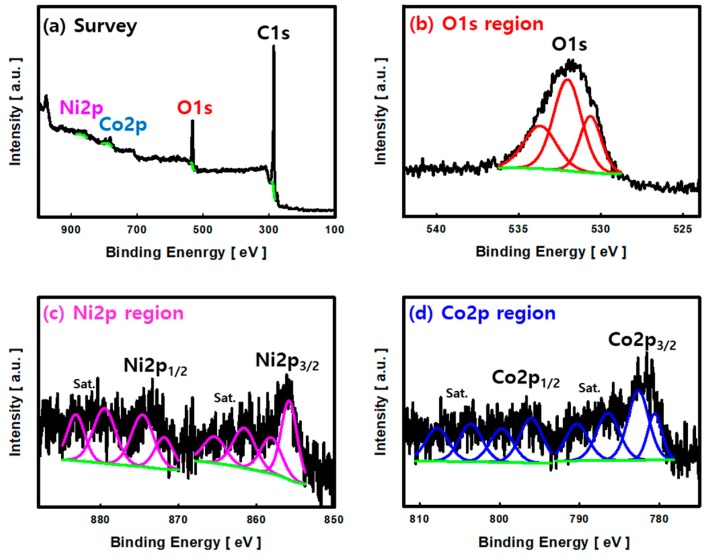
(**a**) wide-scan XPS spectrum, (**b**) O1s, (**c**) Ni2p and (**d**) Co2p XPS spectra of the NCOCC-55 prepared by PiLP.

**Figure 5 nanomaterials-10-00061-f005:**
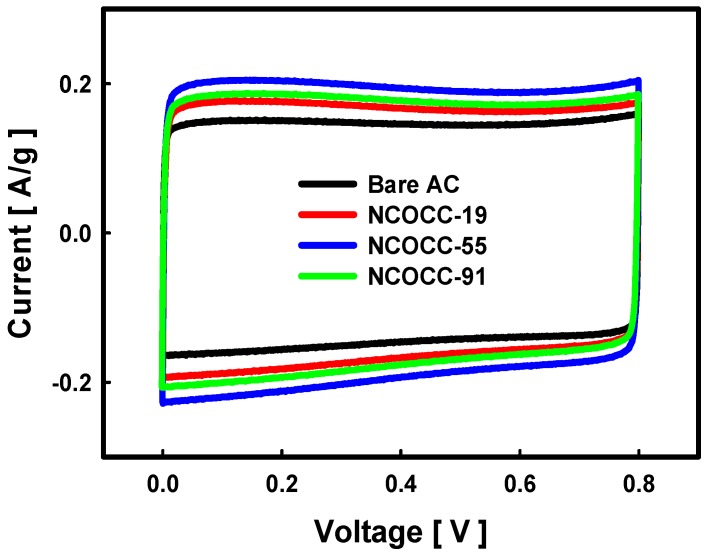
Cyclic voltammograms of NCOCCs electrode prepared by PiLP with different precursor combinations.

**Figure 6 nanomaterials-10-00061-f006:**
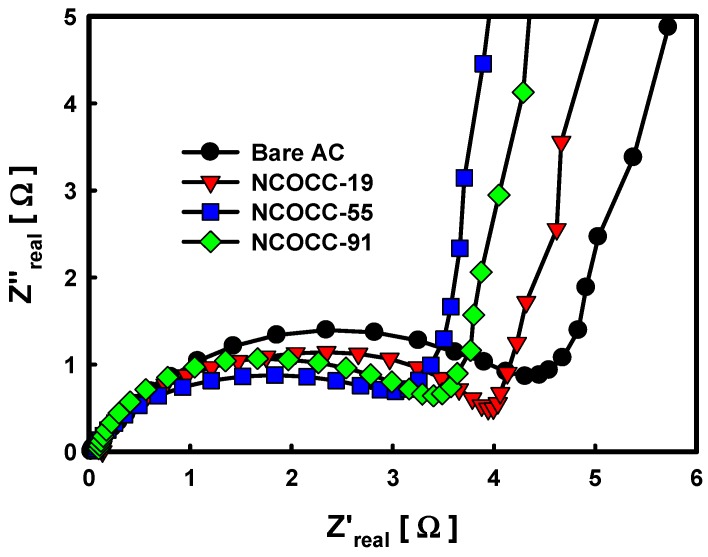
Electrochemical impedance spectroscopy (Nyquist plots) of NCOCCs electrode prepared by PiLP with different precursor combinations.

**Figure 7 nanomaterials-10-00061-f007:**
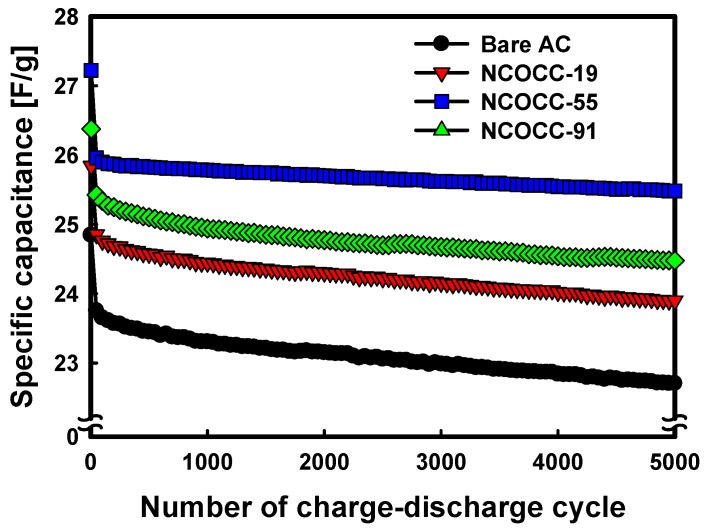
Specific capacitance changed due to charge-discharge during 5,000 cycles of bare AC and NCOCCs.

**Table 1 nanomaterials-10-00061-t001:** Chemical composition of the bare AC and as-prepared composite using PiLP.

Samples	Ni:Co Concentration [mM]	Carbon		Oxygen		Nickel		Cobalt	
		Wt %	At %	Wt %	At %	Wt %	At %	Wt %	At %
Bare AC	**0:0**	96.98	97.71	3.02	2.29	0.00	0.00	0.00	0.00
NCOCC-19	**1:9**	95.03	96.87	3.75	2.87	0.18	0.04	1.04	0.22
NCOCC-55	**5:5**	94.48	96.63	3.97	3.05	0.83	0.17	0.72	0.15
NCOCC-91	**9:1**	94.82	96.79	3.81	2.93	1.24	0.26	0.12	0.02

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
