# Peer review of "Facile Preparation of Ni-Co Bimetallic Oxide/Activated Carbon Composites Using the Plasma in Liquid Process for Supercapacitor Electrode Applications"

_nanomaterials, 2019, doi:10.3390/nano10010061_

Round 1
Reviewer 1 Report
The authors present a new, one-step, facile synthesis of the Ni-Co oxide/activated carbon composite. The composition, structure and properties of the material have been well characterized. However, the paper is poor technically.
Several expressions are very misleading. Just one example: “…new composites doped with transition metals on activated carbon … were synthesized” The abstract and the conclusions section are very similar. What is the major achievement of the work? The doping process results in a rather minor improvement in the material capacity. The vertical axis scale in Figure 7 practically does not exist. There is something wrong with ref. 23.
Author Response
Reviewer #1: The authors present a new, one-step, facile synthesis of the Ni-Co oxide/activated carbon composite. The composition, structure and properties of the material have been well characterized. However, the paper is poor technically.
Several expressions are very misleading. Just one example: “…new composites doped with transition metals on activated carbon … were synthesized”
[Response] We have carefully revised the text, including the points you have indicated.
The abstract and the conclusions section are very similar. What is the major achievement of the work?
[Response] Both the abstract and conclusion sections have been revised, and the main achievements of this study are presented in the abstract and conclusions.
The doping process results in a rather minor improvement in the material capacity.
[Response] This study improves the performance of supercapacitor as electrode by precipitating Ni-Co bimetallic nanoparticles on activated carbon. PiLP, a one-step facile process, was used for the precipitation of Ni-Co bimetallic nanoparticles. The effects of precursor concentrations of Ni and Co on the electrochemical properties of the electrodes of supercapacitor were investigated. The results showed the best performance when using the same molar ratio precursors, and these results are presented in detail.
The vertical axis scale in Figure 7 practically does not exist.
[Response] Revised the interval of the vertical axis scale in Figure 7.
There is something wrong with ref. 23.
[Response] As recommended, revised ref. 23 and carefully revised other references.
Reviewer 2 Report
In this manuscript Ni-Co bimetallic oxide/activated carbon composites were prepared in a single step using a plasma in a liquid process and applied to supercapacitor electrodes. Basically this work should be interesting and will be attractive for broad material scientists and chemical engineering audience. Unfortunately, especially the electrochemical performance of NCOCCs requires a major revision.
Remarks:
1. Introduction section: Some other transition metal oxide materials, such as Fe3O4 and Fe2O3/MnO2 can be referred and sited as well (DOI: 10.1149/2.1161613jes and 10.1021/jp4039573).
2. It is not clear, how the electrochemical measurement cell was assembled? More detail explanation is missing.
3. Cyclic voltammograms of NCOCCs electrode prepared by PiLP with different precursor combinations are quite similar (Figure 5). What is the reproducibility of the different parallel synthesis based measurements?
4. Figure 6 (Nyquist plots) needs more detail explanation. Why the measurements were taken in the range of 1 Hz to 300 kHz? The electrochemical impedance is usually measured to 1 mHz or 5 mHz to obtain equilibrium capacitance values.
Author Response
Reviewer #2: In this manuscript Ni-Co bimetallic oxide/activated carbon composites were prepared in a single step using a plasma in a liquid process and applied to supercapacitor electrodes. Basically this work should be interesting and will be attractive for broad material scientists and chemical engineering audience. Unfortunately, especially the electrochemical performance of NCOCCs requires a major revision.
Introduction section: Some other transition metal oxide materials, such as Fe3O4 and Fe2O3/MnO2 can be referred and sited as well (DOI: 10.1149/2.1161613jes and 10.1021/jp4039573).
[Response] As recommended, added reference and carefully revised other references.
It is not clear, how the electrochemical measurement cell was assembled? More detail explanation is missing.
[Response] Section 2.3 Electrochemical measurement section supplements the preparing details of the coin cell used for the measurement.
Cyclic voltammograms of NCOCCs electrode prepared by PiLP with different precursor combinations are quite similar (Figure 5). What is the reproducibility of the different parallel synthesis based measurements?
[Response] In general, the specific capacitance of the electrode material is measured over an area of cyclic voltammograms. As shown in Figure 5, the C-V curves of AC and NCOCCs are rectangular, but the area of C-V curves of NCOCCs is increased compared to AC. In addition, it was confirmed that the composition change of Ni-Co bimetallic oxide nanoparticles on the AC surface could affect the C-V curve area. In Table 1, it can be seen that the content of bimetal oxide nanoparticles on the AC surface is as low as 1.5% by weight or less. It is thought that C-V curves of similar shape and size are shown due to the substantial difference in content. On the other hand, the difference in the area of the C-V curve represented by the content of the particles was evident, and this difference influenced the specific capacitance. In this measurement, several measurement values were averaged and presented.
Figure 6 (Nyquist plots) needs more detail explanation. Why the measurements were taken in the range of 1 Hz to 300 kHz? The electrochemical impedance is usually measured to 1 mHz or 5 mHz to obtain equilibrium capacitance values.
[Response] The description of Figure 6 is further described and the amendments to the impedance measurement conditions have been revised (performed from 10 mHz to 300 kHz). This study evaluated the effects of Ni-Co bimetallic oxide nanoparticles to improve the electrochemical properties of commonly used AC, and refers to the conditions applied to commercial carbonaceous materials when measuring electrochemical properties including impedance.
Round 2
Reviewer 1 Report
The correction of old ref. 23 was not successful. Now it carries number 25, but it is almost identical to ref 26.
I am satisfied with other replies to my comments.
Author Response
Dear reviewers
Thank you for allowing the opportunity to respond to reviewers’ comments regarding. The manuscript was revised according to the reviewer’s comments. Changes made in response to the comments are described below.
Reviewer #1: The correction of old ref. 23 was not successful. Now it carries number 25, but it is almost identical to ref 26.
[Response] As recommended, revised reference number 25 correctly as follows:
Lee, H.; Park, I.S.; Bang, H.J.; Park, Y.K.; Kim, H.; Ha, H.H.; Kim, B.J.; Jung, S.C. Fabrication of Gd-La codoped TiO2 composite via a liquid phase plasma method and its application as visible-light photocatalysts. Appl. Surf. Sci. 2019, 471, 893-899.
Reviewer 2 Report
One suggested reference (DOI: 10.1149/2.1161613jes) is not included in the introduction section.
Author Response
Dear reviewers
Thank you for allowing the opportunity to respond to reviewers’ comments regarding. The manuscript was revised according to the reviewer’s comments. Changes made in response to the comments are described below.
Reviewer #2: One suggested reference (DOI: 10.1149/2.1161613jes) is not included in the introduction section.
[Response] As recommended, in reference 8, we added the following references:
Eskusson, J.; Rauwel, P.; Nerut, J.; Jänes, A. A Hybrid Capacitor Based on Fe3O4-Graphene Nanocomposite/Few-Layer Graphene in Different Aqueous Electrolytes, J. Electrochem. Soc. 2016, 163, A2768-A2775.